# Can In-Line Iodine Value Predictions (NitFom^TM^) Be Used for Early Classification of Pork Belly Firmness?

**DOI:** 10.3390/foods11020148

**Published:** 2022-01-06

**Authors:** Stephanie Lam, Bethany Uttaro, Benjamin M. Bohrer, Marcio Duarte, Manuel Juárez

**Affiliations:** 1Lacombe Research and Development Centre, Agriculture and Agri-Food Canada, Lacombe, AB T4L 1W1, Canada; stephanie.lam@agr.gc.ca (S.L.); bethany.uttaro@agr.gc.ca (B.U.); 2Department of Animal Sciences, Ohio State University, Columbus, OH 43210, USA; bohrer.13@osu.edu; 3Animal Biosciences Department, University of Guelph, Guelph, ON N1G 2W1, Canada; mduarte@uoguelph.ca

**Keywords:** belly, fatty acid, firmness, iodine value (IV), pork, NitFom

## Abstract

Commercial technologies for assessing meat quality may be useful for performing early in-line belly firmness classification. This study used 207 pork carcasses to measure predicted iodine value (IV) at the clear plate region of the carcass with an in-line near-infrared probe (NitFom^TM^), calculated IV of belly fat using wet chemistry methods, determined the belly bend angle (an objective method to measure belly firmness), and took dimensional belly measurements. A regression analysis revealed that NitFom^TM^ predicted IV (*R*^2^ = 0.40) and belly fat calculated IV (*R*^2^ = 0.52) separately contributed to the partial variation of belly bend angle. By testing different NitFom^TM^ IV classification thresholds, classifying soft bellies in the 15th percentile resulted in 5.31% false negatives, 5.31% false positives, and 89.38% correctly classified soft and firm bellies. Similar results were observed when the classification was based on belly fat IV calculated from chemically analyzed fatty acid composition. By reducing the level of stringency on the percentile of the classification threshold, an increase in false positives and decrease in false negatives was observed. This study suggests the IV predicted using the NitFom^TM^ may be useful for early in-line presorting of carcasses based on expected belly firmness, which could optimize profitability by allocating carcasses to specific cutout specifications.

## 1. Introduction

Pork exports make a substantial contribution to the Canadian economy, with pork belly primal cuts increasing by 30% in value since 2015 (https://www.cpc-ccp.com/statistical-information (accessed on 1 November 2021)). The belly primal cut is sorted for different markets based on quality attributes, such as firmness, in which desirable or high-quality bellies are described as firm, and undesirable or low-quality bellies are described as soft, wet, and floppy. This opens opportunities for increasing Canadian pork export profitability, as high-quality bellies that meet the expectations of premium export markets can achieve higher prices (15 to 20% higher) compared to commodity primals.

Belly firmness is known to be partially influenced by the ratio of unsaturated and saturated fatty acids (FA) in the subcutaneous and seam fat depots, where a higher ratio indicates softer fat, and therefore, softer bellies [1]. Current methods applied in the industry to assess belly firmness include subjective scoring (i.e., how easily the belly flops or folds, oiliness, and finger depression of the fat) which are susceptible to human error, poor scoring guidelines, and inability to account for other potential characteristics influencing firmness. Despite the economic importance of belly quality, the various environmental and physiological factors that impact belly firmness or softness are not fully understood [1].

Alternative objective methods have been used in research to better understand this attribute, including the belly bar bend (central suspension of the belly over a horizontal bar for various time lengths and measuring either the distance between belly ends or the angle created by the bending [2,3,4]), dual-energy X-ray absorptiometry (DEXA) [5], and dimensional measurements of the belly primal cut [6]. More recently, an objective method was developed to measure the belly bend angle, in which the caudal portion of the primal cut is extended off a conveyor belt positioned at 30° and belly bend angle is measured using image analysis [6]. The latter method allows for a highly indicative measure of belly firmness which is fast, nondestructive, and low in cost. Despite the objective methods available in research, currently, no in-line objective method exists in industry to assess belly firmness. Additionally, while current precarcass processing incentives are in place for producers, no postprocessing incentives exist for primal cut quality.

A commonly used objective method to assess fat composition in both the industry and research settings is calculated iodine value (IV), which represents a ratio of the number of unsaturated FA to saturated FA present in the adipose tissue. Iodine value can be calculated with established equations after a chemical analysis has been conducted in the laboratory to identify specific fatty acids. Recently, near-infrared (NIR) spectroscopy technologies have been established to more rapidly predict IV in the plant setting [5]. Due to the biochemical structure of unsaturated FA, a higher IV measurement in pork primal cuts is less favorable, as it indicates softer fat. Commercially available instruments to assess IV in industry includes the NitFom^TM^ (Frontmatec A/S, Smoerum, Denmark), used in the shoulder area of the hot carcass, which is an in-line NIR spectroscopy probe that may have the potential to classify belly primals into different firmness categories and allow for early in-line sorting of bellies. Previous studies have categorized belly firmness using NIR spectroscopy [7]. Additionally, measuring IV in the shoulder via research NIRS devices has been recently reported to have the potential to predict specific belly softness traits, including belly bar bend angle and subjective softness scores of the pork belly [8]. Despite these studies, a rapid in-line system to classify belly firmness objectively has not been assessed for application in the industry. The objective of this study was to evaluate the performance of IV predicted by a commercial device (Nitfom^TM^) on hot carcasses for the early classification of carcasses for pork belly firmness based on the magnitude the cold belly bends.

## 2. Materials and Methods

### 2.1. Carcass and NitFom^TM^ Predicted IV Measurements

Data were collected in a commercial pork processing facility (Ontario, Canada) from 207 pigs originating from various pork producers, which were slaughtered on nine midweek dates scheduled over a 12 week period, in order to generate variability. Pigs were slaughtered under the inspection of the Canadian Food Inspection Agency (CFIA) and slaughter protocols were in accordance with the standard operating procedures of the commercial plant which included CO_2_ stunning of the pigs in groups of six to eight pigs and conventional chilling procedures (temperatures of −4 °C to 2 °C).

Iodine value was predicted from the hot carcass, using the NitFom™ NIT (near infrared transmission) technology, the details and measurements of which are previously described [9,10,11]. Briefly, the NitFom™ probe was inserted approximately 4 cm into the dorsal portion of the shoulder, above the shoulder blade immediately before chilling (approximately 45 min *post-mortem*). Readings were collected at small incremental distances along the probe ejection path (3 cm), which was then used for the algorithm for prediction of IV. Prediction of IV was a result of the direct conversion from three-dimensional data comprised of selected wavelengths from various absorption depths. Twenty-four hours *post-mortem*, carcasses were commercially fabricated according to commercial cutting specifications based on the North American Meat Institute’s Meat Buyer’s Guide [12]. The raw pork belly primal cuts (skin-on, bone-in) were collected after being passed under an in-line belly roller which flattens the cut for easier handling further down the processing line, and were used for conducting the objective, dimensional, and belly bend angle measurements.

### 2.2. Belly Bend Angle and Dimensional Measurements

Objective, dimensional, and belly bend angle measurements were described and illustrated in a diagram in a prior study [6]. Briefly, in this study, belly bend angle (°) was measured from a video recording of the belly bending over the end of a nonstop conveyer with a 30° incline, moving at approximately 24 cm/sec. An increased bend or decreased angle indicates softer bellies, and was measured using image analysis. Additional dimensional measurements comprised belly length (cm) as well as multiple measurements on the loin|belly separation site: side lean thickness (mm) and side fat thickness (mm) at the thickest lean point of the exposed *latissimus dorsi*, total side thickness with and without ribs near the midbelly (mm) and total thickness at or near the bend site (mm).

### 2.3. Calculated IV Measurement

Fat samples were collected from the outer layer of the posterior belly subcutaneous fat and stored at −80 °C until FA analyses. Fatty acid methyl esters (FAME) were prepared according to Jenkins [13] with modifications. A Bruker Scion 436 GC with CP-8400 autosampler, 1079 injector, and flame ionization detector equipped with a Phenomenex Zebron ZB-WAXplus capillary GC column (30 m length × 0.25 mm ID, 0.25 μm film thickness) was used in this study. Scion Instruments Compass CDS Version 4.0.1 software was used to control the gas chromatograph and capture and process chromatographic data. NU-CHEK-PREP GLC reference standard 897 was used for the identification of FAME. Based on the FA composition, the IV of fat was calculated through the equation described by Collison and American Oil Chemists [14] (where square brackets indicate the proportion of an individual FA (% of total FA)):IV = [16:1] × 0.95 + [18:1] × 0.86 + [18:2] × 1.732 + [18:3] × 2.616 + [20:1] × 0.785 + [22:1] × 0.723

### 2.4. Statistical Analysis

Statistical analysis was carried out with SAS (version 9.4, SAS Institute Inc., Cary, NC, USA, 2013). Descriptive statistics of all objective and dimensional measurements were performed using PROC MEANS (Table 1). Stepwise regression models were constructed to determine the prediction performance based on *R*^2^ values of the objective and dimensional measurement values to predict belly bend angle using PROC REG (Table 2). To test the performance of the NitFom^TM^ on classifying belly softness and firmness, quality systems with different percentiles to classify bellies were developed for the predicted IV using the NitFom^TM^, the calculated IV using the belly fat, and belly bend angle. The predicted and calculated IV were categorized with three different minimum quality thresholds defined at the 15th (Predicted IV 15 or Calculated IV 15), 20th (Predicted IV 20 or Calculated IV 20), and 25th (Predicted IV 25 or Calculated IV 25) percentile of the actual values for each trait, meaning values where 85, 80 and 75% of samples would meet the minimum classification threshold of being classified as ‘firm’. The sorted bellies based on the classification thresholds were defined as ‘SOFT’ or ‘FIRM’ quality classes based on belly bend angle, predicted IV, or calculated IV. ‘Predicted IV 15’ represented the threshold classification where the highest 15% of NitFom™ predicted IV were classified as SOFT and the remaining were classified as FIRM. Similar classification thresholds were also developed for the 20 and 25% percentiles. These thresholds were tested against the belly bend angle thresholds, which had a minimum quality threshold defined at the 15th (Angle 15) percentile. ‘Angle 15’ represented the threshold where the lowest 15% of angle values were classified as SOFT and the remaining were classified at FIRM. The latter classification represented the reference method, which best indicated actual belly firmness or softness. False positives were defined as a belly which was classified as SOFT in the IV thresholds and FIRM in the Angle thresholds. False negatives were considered as FIRM in the IV quality system and SOFT in the angle classification thresholds.

## 3. Results and Discussion

### 3.1. Prediction of Belly Firmness

The descriptive statistics of the belly measurements, which were in agreement with previous studies on pork belly primal traits, are reported in Table 1. The predicted IV values using the NitFom™ were similar to those in a previous study which reported an average predicted IV using the NitFom™ of 70.72 ± 6.95 [11]. For belly bend angle, Uttaro et al. [6] reported an average angle of 118.0° for flattened bellies at time 0 with a conveyer angle matching the present study, but moving at a speed of 6.8 cm/sec. Similar dimensional measurements of the belly primal cut have been previously reported and were within a similar range to those reported in this study (belly length 70.6 ± 2.6 cm, width 26.3 ± 1.7 cm, and thickness 6.0 ± 1.1 cm) [1]. The pigs used in this study originated from different farms with the aim of producing variability in IV measurements.

An evaluation of cost-efficient as well as effective, objective approaches to classify belly primal cuts for a variety of quality markets in the pork industry is warranted. Prior studies have evaluated economically important pork quality and carcass traits using predicted IV with the NitFom™ [10,11,15]; however, no studies have assessed pork belly primal quality attributes. This study evaluated the performance of predicted IV (using the NitFom™ probe when inserted into the dorsal portion of the shoulder, above the shoulder blade) using classification thresholds at different levels of stringency, which assessed its potential to correctly classify pork belly primal firmness.

Alternative technologies have been previously studied to assess pork belly quality attributes, including NIR spectroscopy [1,5,8]. A prior study used NIR spectroscopy to predict subjective scores and belly flop angle for lean, intramuscular fat, and fat layers, and revealed prediction accuracies ranging from *R*^2^ = 0.727 to 0.999 for the calibration set and *R*^2^ = 0.436 to 0.715 for the validation set [5]. By including DEXA scans (an alternative objective method for determining cut composition) with NIR of lean and subcutaneous fat layers to predict subjective scores and belly flop angle, results revealed a similar *R*^2^ range of 0.623 to 0.999 for the calibration set, and *R*^2^ range of 0.497 to 0.718 for the validation set [5]. Many studies have used belly flop angle as an assessment for belly firmness, which is a firmness measurement conducted with cut-out specifications (no ribs) and bend location (midbelly), different from the conveyer belly bend measurement used in this study. This demonstrates the unknown variability in belly firmness characteristics, and highlights the multiple factors contributing to belly firmness [1].

Previous studies have conducted partial least squares regression (PLSR) with pork shoulder fat NIR spectra to predict belly flop angle and subjective belly scores, revealing a *R*^2^ range of 80.5 to 90.8% [8]. The latter higher performance, using NIR spectroscopy over a large area of the outer layer of shoulder fat at the midcarcass split, compared to the predicted IV using the NitFom™ in the present study, is reasonable, as the data in this study were dependent on a single point penetrative measure averaged over multiple layers of shoulder fat, which may have reduced the ability to predict belly bend angle. Using PLSR to analyze full NIR spectroscopy data allows for capturing more information compared to the single value output of the NitFom™ IV, as the NitFom™ IV is based on selectively extracted transmission data of at specific wavelengths to optimize speed of analysis. However, despite the high prediction accuracy and performance using data from the full spectrum, the employment of NIR spectra coupled with manually conducted PLSR analysis is currently impractical for industry use due to the speed of analysis and cost. Despite the high speed of IV prediction by the NitFom™, the *R*^2^ may be compromised compared to the study by Lam et al. [8] because the prediction calculation by the NitFom^TM^ may not capture important secondary wavelengths that may be contributing to the variation in belly firmness. Overall, the speed of analysis and economics must be considered when evaluating the best approach for research or industry objectives. Commercial instruments such as the NitFom™ uses NIT spectroscopy to calculate a single average IV of the fat along the path the tip traverses, as a carcass quality measure based on a prediction of the saturated and unsaturated FA ratio. As the NitFom™ IV is readily used in the commercial setting at line-speed to indicate overall carcass quality, and particularly fat quality, this offers the potential for this tool to indirectly measure other primal cut fat-based quality attributes, as well as complex attributes affected by multiple different factors, such as belly firmness. A validation for this measure includes the chemically analyzed IV of a fat sample. This study reported a 15-unit difference between the predicted IV and calculated IV (Table 1). This may be explained by the moderate regression coefficient between calculated IV of belly fat and predicted IV using the NitFom^TM^, which was 0.41. The differences observed between the classification threshold comparisons for each IV method, may also be explained by the NitFom™ company performance test reporting an accuracy of 0.56 for monounsaturated FA prediction and 0.46 for 18:1 (9) FA prediction (www.frontmatec.com/media/3874/frontmatec-instruments-nitfom-_technical-note_web.pdf (accessed on 1 November 2021)). Furthermore, when assessing the regression of the IV methods (predicted IV with the NitFom™ and calculated IV using wet chemistry) with belly bend angle (Figure 1), a relationship between both IV methods was observed. Some extreme values for predicted IV (>85) were also observed (Figure 1), which may be explained by the belly primal cuts originating from different management systems where pigs were fed different diets, as these data points were associated with a specific slaughter group.

The total contribution of each variable to the variation of belly bend angle using either predicted IV and dimensional measurements or calculated IV and dimensional measurements is presented in Table 2. The NitFom™ IV contributed to 40% of the variation of belly bend angle. By including length (model *R*^2^ = 0.46), side lean (model *R*^2^ = 0.49) and total thickness (model *R*^2^ = 0.50) measurements, the total contribution to the variation of belly bend angle increased slightly. Similarly, when assessing the contribution of calculated IV and dimensional measurements to variability in belly bend angle, calculated IV contributed to approximately half (model *R*^2^ = 0.52) of the variation of belly bend angle, while side fat (model *R*^2^ = 0.56) and length (model *R*^2^ = 0.57) accounted for slightly more when added to the model. This also highlights that approximately 50% of belly bend angle variation is still unknown.

As IV is based on FA composition, and different fatty acids exhibit different rheological and structural properties [16], it is reasonable that the predicted IV revealed a high contribution to belly bend variation. Despite this, predicted IV only contributed approximately 40% to differences in bend among bellies. Although adding dimensional measures contributed a small amount to explaining the variation of belly bend angle when assessed with the predicted IV in this study, prior studies have shown a larger contribution of belly thickness to belly bend [1]. In addition, for future studies with higher variation in belly dimensional properties, difference in belly bend angle may be better captured. These results suggest that the NitFom^TM^ may be useful for partial predictions of belly firmness; however, other environmental and physiological factors, and their interactions, that influence belly firmness should also be considered, such as, belly temperature, dimensional properties, lean and fat composition and ratio, muscle pH, management systems, fabrication specifications, and time *post-mortem* [1]. Additionally, dimensional properties such as belly thickness should be further studied due to the lack of variability for dimensional properties in the current study and the evidence of previous studies reporting belly thickness contributing to 33% of the variability in belly firmness [13]. This highlights the nature of the complex trait, belly firmness.

### 3.2. Classification of Belly Firmness

Potential belly firmness and softness classification systems for the industry may be possible, since belly firmness variation is partially explained by predicted IV, as reported in this study. Industry thresholds for payments to producers based on fat quality, as represented by IV, range between maximum values of 70 and 75 [17,18]. In this study, the minimum classification thresholds for identifying the softest bellies were 15, 20 and 25% of belly bend angles starting with the smallest angles, identified as Angle 15, 20 and 25. This resulted in a predicted IV cut-off at ≥74, 72 and 71, and calculated IV cut-off at ≥55.40, 54.90 and 54.63, respectively. The classification threshold of 15% for belly bend angle resulted in an angle cut-off at ≤151.10°. Figure 2 illustrates the frequency of false positives and false negatives when comparing the IV classification threshold performance against Angle 15. The smallest frequency of falsely classified bellies was observed when using the most stringent classification threshold of the 15th percentile. Specifically, using predicted IV 15 resulted in 89.38% correctly classified soft and firm bellies compared to Angle 15 (Figure 2a). Similarly, when comparing the calculated IV classification threshold performance against Angle 15, a comparable accuracy rate of 89.27% was observed when using Belly fat IV 15 (Figure 2b). When reducing the stringency level of both classification thresholds, an increase in false positives and a decrease in false negatives was observed. This reveals that an increase in stringency level leads to fewer bellies being identified as soft when they should be classified as firm, and more being identified as firm when they should be classified as soft. This suggests that as stringency increases, packers will be less likely to lose truly firm bellies (based on the belly bend angle method).

As different premium markets have different requirements for size and trimming of carcass primal cuts, these results suggest an industry-applicable opportunity for presorting belly primal cuts for different markets prior to the trimming step by identifying intact bellies that meet the firmness requirements of a specific market. When assessing the threshold classifications, belly classification was successful at close to 90% for sorting soft bellies at the 15th percentile; however, false positives and false negatives were present. Despite this, the results showed that the classification threshold system using an IV predicted using the NitFom^TM^ probe (inserted into the dorsal portion of the shoulder above the shoulder blade), presented similar results for sorting belly firmness and softness, compared to the calculated IV from belly fat; however, a secondary validation is needed due to the presence of false positives.

By applying this approach in industry, processors may sort and exclude the softest belly primal cuts, which are often considered low quality, and thereby guarantee a standard of belly firmness for premium markets. The soft belly primal cuts will also have an opportunity to be accurately sorted into specific end product markets (i.e., pancetta, low grade bacon, or raw-product markets). This method would also allow for undamaged and unhandled belly primal cuts, as the belly would not need to be probed by the NitFom^TM^. In addition, classification early in the production line allows for appropriate rib removal and trimming for the bellies to be properly sorted into a market to maximize profitability. Processors may increase the level of stringency of classification, but this will also increase the probability of losing high quality bellies while still not achieving 100% accuracy due to the complexity and multifactorial nature of belly firmness.

## 4. Conclusions

This study describes the potential of an in-line near-infrared probe (NitFom^TM^) inserted into the dorsal portion of the shoulder, above the shoulder blade, to classify carcasses based on pork belly quality. Using classification thresholds may assist with early in-line sorting of carcasses and primal cuts for premium markets, resulting in higher belly primal cut value and economic return for processors. This would allow packers to assign different cutout specifications to individual carcasses based on the expected quality of the primals and buyers’ requirements. Further studies could include a secondary validation method to address the presence of false positives.

## Figures and Tables

**Figure 1 foods-11-00148-f001:**
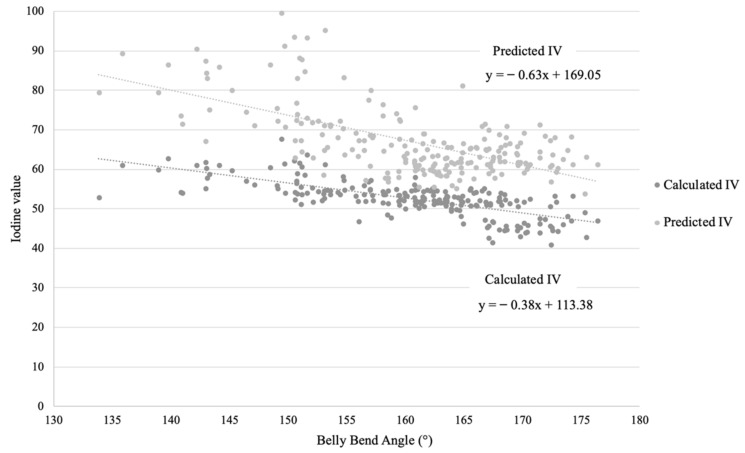
Relationships of predicted (NitFom^TM^) and calculated IV with belly bend angle.

**Figure 2 foods-11-00148-f002:**
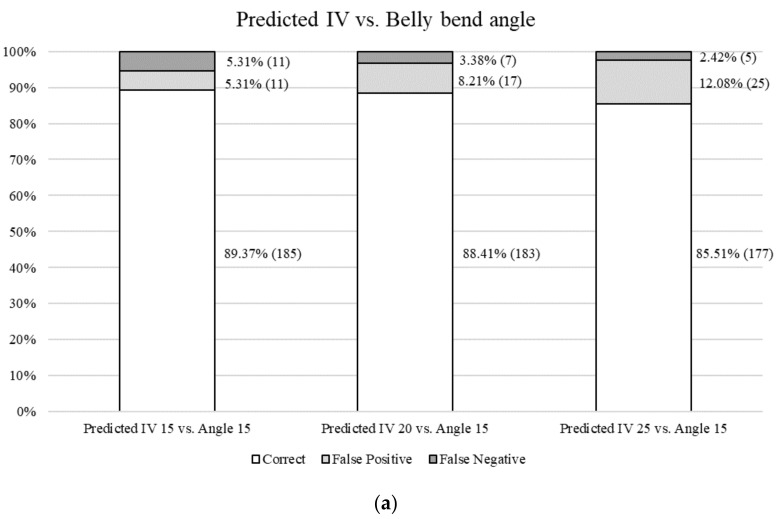
Comparison of levels of stringency for SOFT and FIRM classification thresholds (softest 15, 20 and 25%) for (**a**) predicted iodine value (IV) using an in-line near-infrared probe (NitFom™) inserted into the dorsal portion of the shoulder, above the shoulder blade and (**b**) calculated IV using wet chemistry methods, in comparison to the belly bend angle classi-fication threshold (softest 15%).

**Table 1 foods-11-00148-t001:** Descriptive statistics of belly measurements.

	Mean	SD	Minimum	Maximum
Belly bend and iodine value (IV)				
Belly bend angle (°)	161	8.58	134	177
NitFom™ predicted IV ^i^	67.0	8.64	53.9	99.7
Belly fat calculated IV ^ii^	52.5	4.51	41.0	67.8
Dimensional measurements				
Length (cm)	70.9	3.48	56.0	79.0
Side lean (mm)	31.3	5.25	19.0	44.0
Side fat (mm)	26.4	7.49	8.00	46.0
Side thickness without ribs (mm)	40.9	7.56	20.0	66.0
Side thickness with ribs (mm)	54.8	7.96	25.0	79.0
Total thickness (mm)	56.6	8.02	41.0	84.0

^i^ Predicted IV using an in-line near-infrared probe (NitFom^TM^) inserted into the dorsal portion of the shoulder, above the shoulder blade. ^ii^ Calculated belly fat IV using wet chemistry methods.

**Table 2 foods-11-00148-t002:** Iodine value (IV) and dimensional measurement contributions to predicting belly bend angle using stepwise regression models.

	Partial *R*^2^ Value	Model *R*^2^ Value	C(p)	*F*-Value	*p*-Value
Predicted IV and dimensional measurements					
NitFom™ predicted IV ^i^	0.40	0.40	51.0	135	<0.001
Length (cm)	0.07	0.46	25.4	24.9	<0.001
Side lean (mm)	0.02	0.49	17.5	9.23	0.003
Total thickness (mm)	0.01	0.50	15.5	3.83	0.051
Calculated IV and dimensional measurements					
Belly fat calculated IV ^ii^	0.52	0.52	33.5	223	<0.001
Side fat (mm)	0.03	0.56	19.4	14.9	0.002
Length (cm)	0.02	0.57	13.2	7.84	0.006

^i^ Predicted IV using an in-line near-infrared probe (NitFom^TM^) inserted into the dorsal portion of the shoulder, above the shoulder blade. ^ii^ Calculated belly fat IV using wet chemistry methods.

## Data Availability

The data presented in this study are available on reasonable request.

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
