# Peer review of "Can In-Line Iodine Value Predictions (NitFomTM) Be Used for Early Classification of Pork Belly Firmness?"

_foods, 2022, doi:10.3390/foods11020148_

Round 1

Reviewer 1 Report

Please do the following

  1. Check for grammatical and typo errors
  2. Your equations in the Figure 1 most of the data are scattered. Hence rework on statistical design of the experiment
  3. Need to include error bars in the graphs
  4. Try to formulate a relation with Calculated IV and Predicted IV
  5. Describe and include iodine content of pork and their relation to what they eat

Author Response

REVIEWER 1

  1. Check for grammatical and typo errors

AU: Thank you for your comments. The manuscript has been reviewed in detail for grammatical and typo errors.

  1. Your equations in the Figure 1 most of the data are scattered. Hence rework on statistical design of the experiment

AU: Based on the variability in the herds, the scattered data distribution is expected and was intended to produce variability in the study.

  1. Need to include error bars in the graphs

AU: The graph in Figure 2 is a Frequency plot and therefore does not have or require error bars.

  1. Try to formulate a relation with Calculated IV and Predicted IV

AU: The authors have the plotted results of Calculated IV and Predicted IV in Figure 1 with regression coefficients. Additionally, we have added the regression coefficient between Calculated and Predicted IV at L 216-217.

  1. Describe and include iodine content of pork and their relation to what they eat

AU: We reported the calculated belly fat IV and predicted IV using the NitFom. The dietary information was not available for this study, as these were commercial carcasses sampled at a commercial abattoir, and therefore we cannot make inferences about the IV and diet.

Reviewer 2 Report

In the submitted study, an attempt to assess the pork belly firmness through the stepwise regression models was performed based on iodine value (IV) predictions by NIR, wet chemistry and dimensional belly measurements used to predict belly bent angle, which is currently used as a standard method for belly firmness measurement. As pork belly softness is a common issue during carcass cutting and further processing, the early in-line assessment of pork belly firmness, such as the IV prediction by NIR, could bring a significant practical improvement and economic benefits along the pork processing chain. In that sense, the present study is justified and welcomed. Although the obtained R2 in regression models were not particularly high, the authors fairly argued the remaining variability sources, and the proposed possible belly firmness and softness classification system for the industry is well-explained and elaborated.

If available, the information on animals’ genotype and gender (gilts, castrates or boars) should be also included and discussed in the paper, as these factors may have a significant influence on pork fat properties, including the FA composition.

Technical remark – extra spaces in the text should be corrected (e.g. rows 140, 203, 223, 248, 264…).

Author Response

REVIEWER 2

In the submitted study, an attempt to assess the pork belly firmness through the stepwise regression models was performed based on iodine value (IV) predictions by NIR, wet chemistry and dimensional belly measurements used to predict belly bent angle, which is currently used as a standard method for belly firmness measurement. As pork belly softness is a common issue during carcass cutting and further processing, the early in-line assessment of pork belly firmness, such as the IV prediction by NIR, could bring a significant practical improvement and economic benefits along the pork processing chain. In that sense, the present study is justified and welcomed. Although the obtained R2 in regression models were not particularly high, the authors fairly argued the remaining variability sources, and the proposed possible belly firmness and softness classification system for the industry is well-explained and elaborated.

AU: Thank you for your comments.

If available, the information on animals’ genotype and gender (gilts, castrates or boars) should be also included and discussed in the paper, as these factors may have a significant influence on pork fat properties, including the FA composition.

AU: Information on the animal breed or gender was unavailable for this study. In addition, the focus of the study was to test belly firmness classification systems which required variability in the belly bend and IV measurements. Therefore, the variability in belly quality due to other factors such as breed or gender was not important for this study.

Technical remark – extra spaces in the text should be corrected (e.g. rows 140, 203, 223, 248, 264…).

AU: We have corrected the extra spaces throughout the manuscript.

Reviewer 3 Report

The manuscript entitled „Can in-line iodine value predictions (NitFomTM) be used for early classification of pork belly firmness?” in my opinion could be published in Foods.

Comments to the manuscript are presented below:

  1. Can the authors specify the breeds of pigs used in the study, was the possible breed distribution consistent with the breeding stock of the individual breeds on the Canadian market? The question is related to the different size of the carcass and therefore the different size of the bacon.
  2. In the formula for IV there is no sign "x" after [18:2]
  3. Results and Discussion - remove the paragraph immediately after the section title.
  4. Line 212 replace NIT with NIR
  5. Will the withdrawal of belly bend angle <145 results from the statistical calculations (due to their very small number) cause a significant change in the discussed results. According to Fig. 1, it seems that these results have a significant influence on the direction of the curves.

The submitted comments are debatable (or editorial) and do not affect the positive evaluation of the article.

Author Response

REVIEWER 4

  1. Can the authors specify the breeds of pigs used in the study, was the possible breed distribution consistent with the breeding stock of the individual breeds on the Canadian market? The question is related to the different size of the carcass and therefore the different size of the bacon.

AU: Thank you for your comments. We used commercial carcasses sampled at a commercial abattoir throughout multiple regular operation days. This would provide samples representative of commercial pork production in Canada and variability in quality attributes. No live animal information was available for this study.

  1. In the formula for IV there is no sign "x" after [18:2]

AU: Thank you for catching this error. We corrected it in the manuscript.

  1. Results and Discussion - remove the paragraph immediately after the section title.

AU: We removed this description paragraph.

  1. Line 212 replace NIT with NIR

AU: The acronym NIT represents near infrared transmission and was correct.

  1. Will the withdrawal of belly bend angle <145 results from the statistical calculations (due to their very small number) cause a significant change in the discussed results. According to Fig. 1, it seems that these results have a significant influence on the direction of the curves.

AU: To test his, we could remove some of the data, however, that would modify the equations and we do not have reason to do so. We cannot assume the data is wrong and choosing to remove it is not justified. Although the authors agree that a larger dataset would increase the accuracy of the algorithm, the concepts discussed in this study and the conclusions would not change.

The submitted comments are debatable (or editorial) and do not affect the positive evaluation of the article. 
